# Invertebrate Gonadotropin-Releasing Hormone Receptor Signaling and Its Relevant Biological Actions

**DOI:** 10.3390/ijms21228544

**Published:** 2020-11-12

**Authors:** Tsubasa Sakai, Tatsuya Yamamoto, Shin Matsubara, Tsuyoshi Kawada, Honoo Satake

**Affiliations:** Bioorganic Research Institute, Suntory Foundation for Life Sciences, 8-1-1 Seikadai, Seika, Souraku, Kyoto 619-0284, Japan; sakai@sunbor.or.jp (T.S.); yamamoto@sunbor.or.jp (T.Y.); matsubara@sunbor.or.jp (S.M.); kawada@sunbor.or.jp (T.K.)

**Keywords:** gonadotropin-releasing hormone receptors, cell signaling, invertebrates, evolution, biological functions

## Abstract

Gonadotropin-releasing hormones (GnRHs) play pivotal roles in reproduction via the hypothalamus-pituitary-gonad axis (HPG axis) in vertebrates. GnRHs and their receptors (GnRHRs) are also conserved in invertebrates lacking the HPG axis, indicating that invertebrate GnRHs do not serve as “gonadotropin-releasing factors” but, rather, function as neuropeptides that directly regulate target tissues. All vertebrate and urochordate GnRHs comprise 10 amino acids, whereas amphioxus, echinoderm, and protostome GnRH-like peptides are 11- or 12-residue peptides. Intracellular calcium mobilization is the major second messenger for GnRH signaling in cephalochordates, echinoderms, and protostomes, while urochordate GnRHRs also stimulate cAMP production pathways. Moreover, the ligand-specific modulation of signal transduction via heterodimerization between GnRHR paralogs indicates species-specific evolution in *Ciona intestinalis*. The characterization of authentic or putative invertebrate GnRHRs in various tissues and their in vitro and in vivo activities indicate that invertebrate GnRHs are responsible for the regulation of both reproductive and nonreproductive functions. In this review, we examine our current understanding of and perspectives on the primary sequences, tissue distribution of mRNA expression, signal transduction, and biological functions of invertebrate GnRHs and their receptors.

## 1. Introduction

Reproduction is one of the most fundamental physiological functions for the continued existence and evolution of an animal. All species have developed finely tuned reproductive systems regulated by a wide variety of factors, including the environment, metabolic and nutritional states, maturity, and endogenous compounds [1]. In vertebrates, the decapeptide gonadotropin-releasing hormone 1 (GnRH1) plays pivotal roles as a major hypothalamic peptide hormone in reproductive functions via the hypothalamus-pituitary-gonad (HPG) axis [2]. GnRHs also act as neuropeptides in diverse nonreproductive functions [2,3,4,5,6]. Most vertebrate GnRHs comprise 10 amino acids and harbor the consensus sequence pyro-Glu^1^-His^2^-Trp^3^-Ser^4^, Gly^6^, and Pro^9^-Gly^10^-amide (Table 1). These common residues are essential for receptor binding and activation [2,7].

GnRHs exhibit their functions via specific interactions with the cognate receptor GnRHR. GnRHRs belong to the class AG-coupled receptor (GPCR) family, and one to three GnRHRs are encoded in vertebrates [2,7,8,9]. GnRHRs are coupled to the Gq protein, which is activated upon binding to GnRH [2,7,8]. An activated Gq protein α subunit allows phospholipase Cβ (PLC) to generate inositol triphosphate (IP_3_), then IP_3_ induces intracellular calcium mobilization [2,7,8]. Thus, IP_3_ and intracellular calcium are major second messengers of GnRH-GnRHR systems. These second messengers activate protein kinase C (PKC), leading to the phosphorylation of mitogen-activated protein kinases (MAPKs), including extracellular signal-regulated protein kinase (ERK)1/2 [8]. GnRHRs are also coupled to Gi and Gs proteins in cell-context or tissue-dependent fashions [2,7,8].

Over the past quarter-century, GnRH-GnRHR signaling systems have also been identified by molecular and in silico cloning or transcriptome analysis in invertebrate deuterostomes and protosomes: urochordates, cephalochordates, echinoderms, crustaceans, mollusks, and annelids (Table 1). These species lack a hypothalamus and pituitary. Absence of the HPG axis and gonadotropins in invertebrates indicates that biological functions of GnRHergic signaling in these species do not include the release of gonadotropins. The evolutionary relationships of the GnRH-related peptides corazonins (CRZs) and adipokinetic hormones (AKHs) have been identified, and the cognate receptors of these structurally related peptides have been documented based on GnRHR, CRZ receptor, and AKH receptor molecular phylogenetic trees [9,10,11,12,13]. There is a recent growing body of reports describing GnRH-GnRHR signaling and the resultant biological effects in invertebrates. These insights will aid investigations into the biological roles of GnRHs in invertebrates and into evolutionary aspects of GnRH-related peptides.

In this review, we summarize our current understanding of the primary sequences, tissue distribution, signaling cascades, and biological functions of invertebrate GnRHs and their receptors.

## 2. Urochordate GnRH Signaling

### 2.1. Urochordate GnRHs

Ascidians are invertebrate deuterostome marine animals belonging to the superphylum vertebrates, chordata, and the phylum urochodata [14]. Their critical phylogenetic position makes them attractive in evolutionary biology as a model for the peptidergic regulation of reproduction [15]. Twelve GnRH peptides have been identified in ascidians (Table 1). tGnRH-1 and -2 (Table 1) were originally identified in the neural tissue of the ascidian *Chelyosoma productum* [16] in the first characterization study conducted on invertebrate GnRHs. Subsequently, ascidian GnRHs were characterized from the cerebral ganglions of the ascidian *Ciona intestinalis* Type A and the closely related species *Ciona savignyi* [17]. The former ascidian produces tGnRH-3 to -8 (Table 1), and the latter generates tGnRH-5 to -9 (Table 1) [17]. tGnRH-10 and -11 (Table 1) from *Halocynthia roretzi* have also been characterized [18]. It is noteworthy that all these ascidian GnRHs conserve the consensus sequence pyro-Glu^1^-His^2^-Trp^3^-Ser^4^ and Pro^9^-Gly^10^-amide of vertebrate GnRHs. However, ascidian GnRH genes encode multiple copies of GnRH sequences in a single precursor, unlike vertebrate and non-ascidian invertebrate GnRH genes, which encode a single GnRH sequence [7]. For instance, *Ci-gnrh-1* encodes tGnRH-3, -5, and -6, whereas tGnRH-4, -7, and -8 sequences are found in another gene, *Ci-gnrh-2* [17]. Likewise, the *H. roretzi* GnRH gene encodes tGnRH-10 and -11 [18]. Intriguingly, the GnRH structurally related peptide Ci-GnRH-X was identified in the central nervous system of *C. intestinalis* as an endogenous antagonist against tGnRH-3, -5, and -6 [19]. Ci-GnRH-X comprises 16 amino acids harboring the consensus sequence pyro-Glu^1^-His^2^-Trp^3^-Ser^4^ and Pro^9^-Gly^10^ and a *C*-terminal Gly-amide (Table 1) [19]. These findings indicate conservation and the species-specific diversification of GnRHs in urochordates. Recently, the precise microscopic observation of the nervous system of transgenic adult *C*. *intestinalis*, which expresses the *Kaede* reporter gene driven by a protein convertase 2 gene promoter, suggested that neuropeptides produced in the cerebral ganglion are secreted to the ovary via neurons of the dorsal strand plexus and/or visceral nerves and regulate the gonads [20]. This observation is consistent with reports on the distribution of GnRH immunoreactive neurons of the dorsal strand plexus from the cerebral ganglion to the gonads along the dorsal strand of *C. intestinalis* [17,21].

### 2.2. Urochordate GnRH Receptors

The four *Ciona* GnRHRs (Ci-GnRHR)-1 to -4 have been identified as being phylogenetically closest to vertebrate GnRH receptors and shown to exhibit characteristic cell signaling involving ligand-receptor selectivity, coupling with multiple G-protein subtypes, and receptor heterodimerization (Figure 1 and Figure 2 and Table 1). The sequences of Ci-GnRHR-1, -2, and -3 harbor a long *C*-terminal tail, whereas a short tail is present in the *C*-terminus of Ci-GnRHR-4 [22,23]. Molecular phylogenetic analyses revealed that Ci-GnRHRs are included in vertebrate GnRHR clades but form an independent cluster in chordate GnRHRs, suggesting that these receptors might have evolved within the *Ciona* species [7,9,15,23,24].

Elevated IP_3_ generation, and the induction of calcium ion mobilization, are typical responses of vertebrate GnRHR activation and were observed only upon the binding of tGnRH-6 to Ci-GnRHR-1 expressed in COS-7 cells [23,24] and in HEK293MSR cells [24,25], respectively. In addition, Ci-GnRHR-1 induces cAMP production in response to tGnRH-6 in these cells [23,24,25]. These findings demonstrated that the tGnRH-6/Ci-GnRHR1 pair, like mammalian GnRH-GnRHR counterparts [2], activate both the PLC-IP_3_ intracellular calcium and adenyl cyclase-cAMP signaling cascades (Table 1).

The induction of calcium mobilization by tGnRH-6 via Ci-GnRHR-1, expressed in HEK293MSR cells, was shown to lead to the translocation of a typical calcium ion-dependent protein kinase, PKCα, from the cytoplasm to the plasma membrane. This is a canonical activation process of PKC [25]. The tGnRH-6-Ci-GnRHR-1 interaction was also found to enhance the translocation of a calcium ion-independent PKC, PKCζ [25], indicating that the tGnRH-6-Ci-GnRHR-1 interaction activates different signaling pathways (Figure 1). Furthermore, tGnRH-5 was shown to enhance the translocation of the calcium ion-independent PKC, PKCζ but not PKCα, although tGnRH-5 fails to elevate the intracellular calcium or cAMP production [25]. We also demonstrated that the activation of both PKCα and PKCζ triggered a canonical signaling process—the phosphorylation of ERK1/2, which belongs to the MAPK family [25]. In combination, tGnRH-5 and -6 regulate multiple signaling cascades via calcium ion-dependent and independent pathways, leading to the phosphorylation of ERK1/2 (Figure 1).

Ci-GnRHR-2 expressed in COS-7 cells [23] or in HEK293MSR cells [24,26] exclusively stimulates cAMP production in response to tGnRH-7, -8, and -6 in this order of potency [23,24,26]. Ci-GnRHR-3 triggers cAMP production in the presence of tGnRH-3 and -5 to a similar extent in a ligand-specific fashion in both cell types [23,24]. Ci-GnRHR-4 elevated neither the intracellular calcium nor the production of cAMP [23,24,25,26,27], suggesting that Ci-GnRHR-4 is a nonfunctional receptor [23]. These ligand-selective activities verified that cAMP is the major second messenger of *Ciona* GnRHergic signaling cascades, in contrast with vertebrate GnRHergic signaling. Furthermore, unique *Ciona* GnRHergic signaling is characterized by a partial antagonistic effect of Ci-GnRH-X on Ci-GnRHR-1 and -3 [19]. Ci-GnRH-X inhibited 10–50% of the induction of intracellular calcium and cAMP production by tGnRH-6 at Ci-GnRHR-1 and cAMP production by tGnRH-3 and -5 via Ci-GnRHR-3 in a dose-dependent manner. In contrast, Ci-GnRHR-2 was not inhibited by Ci-GnRH-X [19]. Partial antagonistic effects have not been detected in any other endogenous peptidergic systems involving GnRHergic systems. These findings also indicate multiple and species-specific GnRH signaling cascades in urochordates.

*Ciona* GnRH signaling is further diverged via multiple heterodimers between Ci-GnRHR-1 and Ci-GnRHR-4 and between Ci-GnRHR-2 and Ci-GnRHR-4 (Figure 1 and Figure 2). Notably, the heterodimerization of Ci-GnRHR-4 with Ci-GnRHR-1 resulted in a 10-fold elevation of intracellular calcium compared with the Ci-GnRHR-1 monomer/homodimer [25]. Moreover, the enhancement of intracellular calcium leads to a more prominent activation of PKCα by tGnRH-6 and PKCζ by tGnRH-5 and -6, followed by the upregulation of ERK phosphorylation (Figure 1C,D) [25]. Ci-GnRHR-4 was also found to heterodimerize with Ci-GnRHR-2 (Figure 2) [26]. Furthermore, the Ci-GnRHR-2/-4 heterodimer decreased cAMP production by 50% in a non-ligand-selective manner by shifting the activation from the Gs protein to Gi protein by Ci-GnRHR-2 compared with the Ci-GnRHR-2 monomer/homodimer [26]. These findings verify that Ci-GnRHR-4 serves as a protomer of Ci-GnRHR heterodimers rather than as a ligand-unidentified orphan GPCR (Figure 2) [7,15,27]. Taken together, these findings indicate urochordate-specific molecular and functional diversity in GnRH signaling systems.

### 2.3. Biological Functions

#### 2.3.1. Reproductive Effects

Ascidians are not endowed with a hypothalamus or a pituitary in the central nervous system, or with gonadotropin genes, and thus, tGnRHs cannot serve as “gonadotropin-releasing hormones” in the HPG axis but, rather, function as neuropeptides that directly regulate target tissues. Tissue distribution of the *gnrh* genes *Ci-gnrh-1* and *-2* and their receptor genes *Ci-gnrhr-1* to *-4* in adult *C. intestinalis* strongly support this notion. RT-PCR analysis of total RNA extracted from the neural complex, testis, ovary, heart, and hepatic organ of adult *C. intestinalis*, and in situ hybridization, revealed that the two *gnrh* genes are exclusively expressed in the neural complex, especially in the cerebral ganglion [28]. In contrast, the four *gnrhr* genes are expressed in various tissues, such as the neural complex, gonad, heart, intestine, endostyle, and branchial sac in adult *Ciona* [23]. These gene expression profiles of tGnRHs and Ci-GnRHRs suggest that urochordate GnRH signaling regulates reproductive and nonreproductive activities. Notably, several studies demonstrated that the administration of tGnRHs to adult urochordates influences the reproductive activities of the ascidian. In *C. intestinalis*, tGnRHs were found to increase the water flow by body contraction and then induce the release of eggs and sperm by the injection of water into the gonaducts, ovary, stomach, and posterior body cavity [17,21]. Furthermore, the colocalization of Ci-GnRHR-4 and Ci-GnRHR-1 or R-2 was detected exclusively in test cells inside the inner follicles of oocytes in the vitellogenic follicle, indicating the involvement of GnRH signaling in oocyte growth via GPCR heterodimerization [25,26,27].

#### 2.3.2. Nonreproductive Effects

Expression of the GnRH genes *Ci-gnrh-1* encoding tGnRH-3, 5, and 6; *Ci-gnrh-2* encoding tGnRH-4, 7, and 8; and four GnRH receptor genes have been reported in the nonreproductive stage of *C. intestinalis* [17,24,29,30]. *Ci-gnrh1* and *Ci-gnrh2* are expressed at the four-cell stage and gastrulation, tailbud, and tail absorption stages of larvae during development [17]. Furthermore, these genes are expressed in the motor ganglion and central nervous systems of larvae, including in papillar neurons, and ascidian metamorphosis is triggered upon adhesion to a substrate [24,30]. All four *Ci-gnrhr* genes are also expressed in the central nervous system of the larva [24,30]. Moreover, *Ci-gnrhr-1* and *-2* are expressed in the adhesive papillae and tail muscle, whereas *Ci-gnrhr-3* is expressed in notochord cells in the larval tail, which is rapidly absorbed during metamorphosis [24]. Such a distribution of gene expression suggests that GnRH signaling regulates the sensory reception, neuronal processing, and muscle functions in *Ciona* larvae [24]. Importantly, our group demonstrated that tGnRH-3 and -5 induce tail absorption and inhibit the growth of adult organs in the larval trunk by arresting cell cycle progression [29]. These findings indicate that tGnRHs play a pivotal role in metamorphosis. Intriguingly, expression of the Ci-GnRH-X gene was detected specifically in the larva, as well as in the adult neural complex [19]. The specific interactions of tGnRH-3 and -5 with Ci-GnRHR-3, and a partial inhibitory effect of Ci-GnRH-X on Ci-GnRHR-3, suggested that the antagonistic effect of Ci-GnRH-X on Ci-GnRHR-3 is likely indicative of a role in the metamorphosis of the larva. Quite recently, gene functional analyses and pharmacological assays demonstrated that gamma-butyric acid (GABA) induces the secretion of GnRH through the metabotropic GABA receptor to initiate *Ciona* metamorphosis [30]. *Ci-gnrh2*-knockout larvae produced by genome editing exhibited failed tail regression, while *Ci-gnrh1*-knockout counterparts exhibited a moderate reduction of tail regression. This study indicated that the GABA-GnRH axis participates in postembryonic development in ascidians [30].

## 3. Cephalochordate GnRH Signaling

### 3.1. Cephalochordate GnRHs

A hidden Markov model for a variety of vertebrate and invertebrate GnRH, AKH, and CRZ propeptides in the cephalochordate amphioxus *Branchiostoma floridae* identified two putative GnRH-like peptides: amphioxus GnRHv (pQEHWQYGHWYa, Table 1) and the invertebrate GnRH- and CRZ-like peptide amphioxus GnRH-like peptide (pQILCARAFTYTHTWa, Table 1) [31]. Synteny analysis of the amphioxus GnRH-like peptides with human GnRH paralogons and limpet AKH indicated the lineage of the ancestral-type GnRH superfamily peptides and its evolutionary relevance to vertebrate GnRHs [31].

### 3.2. Cephalochordate Receptors

Four putative GnRHRs have been cloned from the amphioxus *B. floridae* [32]. Two putative GnRH-like peptides and several peptides were tested for bioactivity in vitro with the four receptors. Amphioxus GnRHR-1 and -2, expressed in COS-7 cells, were activated by vertebrate GnRH1 (pQHWSYGLRPGa) but not by Amph.GnRHv and Amph.GnRH-like peptides. Amphioxus GnRHR-3 was activated exclusively by the Amph.GnRH-like peptide oct-GnRH (pQNYHFSNGWPGa) and by silkworm AKH (pQLTFTSSWa) at physiological concentrations [31,32]. In contrast, no ligands induced IP_3_ accumulation or cAMP stimulation via amphioxus GnRHR-4 [31,32]. This functional assay indicated that Amph.GnRHv, which has a higher sequence similarity to other species’ GnRHs than the AmphGnRH-like peptide, does not interact with any hereto known putative amphioxus GnRHRs [31]. The response of amphioxus GnRHR-4 remains to be investigated, but mismatches between putative amphioxus GnRHRs and cultured cells used for heterologous functional analysis may cause unsuccessful functional expression of the receptor mRNA or degradation of the receptor protein in heterologous expression systems, as seen with other invertebrate GPCRs [33]. Molecular phylogenetic analysis demonstrated that amphioxus GnRHR-1 and -2 are members of the vertebrate GnRHR clade, while amphioxus GnRHR-3 and -4 belong to the CRZ receptor/invertebrate GnRHR clade. Collectively, the sequences of Amph.GnRH-like peptides might reflect the ancestral sequence of CRZ/GnRH or the transition state between CRZ and GnRH [31].

To our knowledge, the detailed tissue distribution of amphioxus GnRHs and their receptors has not been published. Thus, the biological functions of GnRHergic signaling in the amphioxus remain unclear. The identification of authentic ligands for amphioxus GnRHR-1 and -2 and the tissue distribution of amphioxus GnRH and receptor pairs await further investigation.

## 4. Echinoderm GnRH Signaling

### 4.1. Echinoderm GnRHs

To date, two echinoderm GnRH peptides have been identified. The GnRH-like peptide SpGnRHP (Table 1) was identified in the sea urchin *Strongylocentrotus purpuratus* as the first echinoderm GnRH-related peptide [34], followed by the identification of ArGnRH (Table 1) in the starfish *Asterias rubens* [35]. SpGnRHP and ArGnRH comprise 12 amino acids, with Val^2^-His^3^ and Ile^2^-His^3^ inserted in the *N*-terminus of these peptides, respectively (Table 1). Furthermore, the echinoderm GnRHs share several amino acids with urochordate and vertebrate GnRHs and protostome GnRH-like peptides (see Section 4.2), including the *N*-terminal pGlu, His^4^ (corresponding to His^2^ in chordate GnRHs), Gly^8^ (corresponding to Gly^6^ in vertebrate GnRHs), Trp^9^ (corresponding to Trp^7^ in vertebrate GnRHs), and *C*-terminal Pro-Gly-amide (Table 1). SpGnRHP and ArGnRH were recently shown to have weak sequence similarity with CRZ (pQTF(Q/H)YS(R/Q)GW(T/Q)N-amide), an arthropod neuropeptide [9,10,11]. Consequently, it has been presumed that echinoderm GnRHs are included in the GnRH/CRZ family [9,10,11]. In addition, ArGnRH was shown to bind to copper (II) ion and nickel (II) ion with much higher affinity compared with human GnRH [36]. Although the physiological significance of this high-affinity binding remains to be investigated, the *N*-terminus of ArGnRH forms an ATCUN motif copper binding site, in which the third histidine is responsible for the high-affinity binding of copper and nickel [36].

### 4.2. Echinoderm GnRH Receptors

Genome surveys suggested that four GnRH/CRZ-type receptors are encoded in the sea urchin *S. purpuratus* [9], although the functions of these receptors remain to be investigated. These four receptors belong to a separate cluster in the molecular phylogenetic tree of GnRH receptors and CRZ receptors, indicating that these receptors were generated in the sea urchin lineage [9]. The cognate receptor for ArGnRH, ArGnRHR, was identified in the starfish *A. rubens* [35], and ArGnRH was shown to specifically activate intracellular calcium mobilization via ArGnRHR expressed in CHO-K1 cells [35]. Molecular phylogenetic tree analysis demonstrated that ArGnRHR forms a cluster with the aforementioned four sea urchin putative GnRHRs, suggesting that echinoderm GnRHRs were generated in common ancestors of echinoderms [9]. In addition, ligand-receptor signaling for CRZ was also identified in *A. rubens* [35]. In combination, these findings indicate that GnRH-type and CRZ-type signaling pathways function in *A. rubens* [10,11] and represent the first functional characterization of GnRH-type and CRZ-type signaling pathways in an organism.

### 4.3. Biological Effects of Echinoderm GnRH

Of the echinoderm GnRHs identified to date, only the biological effects of ArGnRH have been characterized. ArGnRH was shown to stimulate the contraction of cardiac, stomach, and apical muscles and tube foot preparations in vitro [37], suggesting that ArGnRH participates in regulating locomotion and food digestion. In contrast, no reproductive functions of this neuropeptide have been examined. Investigation of the tissue distribution of ArGnRHR and ArGnRHergic innervation would provide insight into the reproductive roles of ArGnRH.

## 5. Protostome GnRH Signaling

### 5.1. Protostome GnRHs

GnRH-like peptides have been identified over the past fifteen years in protostomes, including mollusks, annelids, and crustaceans (Table 1). The first protostome GnRH, oct-GnRH, was isolated from the central nervous system of the octopus *Octopus vulgaris* [38], followed by the identification of oct-GnRH in other cephalopods such as cuttlefish *Sepia officinalis* and swordtip squid *Loligo edulis* [7,9,39]. Oct-GnRH, like echinoderm GnRHs, is composed of 12 amino acids and conserves amino acids corresponding to pyro-Glu, His^2^, Ser^4^, Gly^6^, Pro^9^, and *C*-terminally amidated Gly^10^ in typical vertebrate GnRHs and tunicate t-GnRH6, -10, and -11 (Table 1) This amino acid sequence identity demonstrates that oct-GnRH shares more common amino acids with vertebrate and tunicate GnRHs compared with the echinoderm GnRHs SpGnRHP and ArGnRH. Oct-GnRH-like peptides [9,40,41,42,43,44] have also been identified in other protostomes, including bivalves, gastropods, annelids, and crustaceans (Table 1). In contrast, GnRH-like peptides have not been identified in insects or nematodes, although CRZs and AKHs, which are believed to have originated from common ancestors of GnRHs, are found in these organisms [7,9,11]. Unlike deuterostome and cephalopod GnRHs, these non-cephalopod protostome GnRHs, except Has-GnRH [40], comprise 11 amino acids and lack the *C*-terminal Gly-amide (Table 1). However, most protostome GnRHs also conserve amino acid residues corresponding to pyro-Glu, His^2^, Ser^4^, Gly^6^, Trp^7^, Pro^9^, and *C*-terminal amidation in vertebrate GnRHs and tunicate tGnRH6, -10, and -11 (Table 1). Such sequence identity and diversity suggest that common ancestral GnRHs might have harbored *C*-terminal Pro-Gly-amide, which is conserved in cephalopods and echinoderms but was lost during the evolutionary processes of protostomes. Consistent with this, the sequence diversity of GnRH showed that ancestral GnRHs might have comprised pQ-H(F/W)S-GW-PGa or pQ-H(F/W)S-GW-a. Subsequent vertebrate and tunicate GnRHs could have arisen via deletion of the two *N*-terminal amino acids and several substitutions of amino acids during the evolutionary process of each species.

### 5.2. Protostome GnRH Receptors

As shown in Table 1, putative protostome GnRHR sequences were detected in several species [9,11,43,45,46], but only three protostome GnRHRs have been shown to interact with the cognate GnRH ligands. The first protostome GnRHR identified was oct-GnRHR in the octopus *O. vulgaris* [47]. Oct-GnRHR expressed in *Xenopus* oocyte activates intracellular calcium mobilization in response to oct-GnRH but not to vertebrate GnRHs. Moreover, an oct-GnRH synthetic analog lacking Asn^2^-Tyr^3^ failed to activate calcium mobilization via oct-GnRHR, whereas a chicken GnRH-II synthetic analog with an Asn-Tyr insertion after position 1 exhibited weak activity [47]. These findings revealed that Asn^2^-Tyr^3^ is a requisite for signaling by oct-GnRHR, suggesting that the two amino acids after position 1 in non-chordate GnRHs are responsible for activating protostome GnRHRs. In the gastropod sea hare (*Aplysia californica*), ap-GnRH was shown to increase IP_3_ accumulation but not cAMP production in ap-GnRHR-expressing *Drosophila* S2 cells [48]. Recently, my-invGnRHR was identified in the bivalve yesso scallop (*Mizuhopecten* (*Patinopecten*) *yessoensis*) [49]. This receptor, transfected into HEK293MSR cells, was shown to trigger intracellular calcium mobilization, but not cAMP production, in response to the cognate ligand py-GnRH [49]. Molecular phylogenetic trees demonstrated that these molluscan GnRHRs are clustered in a separate clade and suggested that protostome GnRHRs might have originated from common ancestors of deuterostome GnRHRs, CRZ receptors, and AKH receptors [9,11,49]. Taken together, these findings support the evolutionary view that the original GnRHRs (or their common ancestors) might have only been capable of activating the PLC-IP_3_-intracellular calcium mobilization signaling cascade, whereas ascidian and vertebrate GnRHRs both conserved the PLC-IP_3_-intracellular calcium mobilization signaling cascade and acquired cAMP production signaling during their evolutionary processes.

### 5.3. Biological Effects of Protostome GnRHs

In vitro experiments showed that oct-GnRH stimulates the contractions of the heart and oviduct, and secretion of testosterone-, progesterone-, and 17β-estradiol-immunoreactive steroids, from the follicle and spermatozoa in *O. vulgaris* [47,50]. These effects are compatible with the expression of *oct-gnrhr* in the heart, oviduct, testis, and ovary [47,50]. The localization of *oct-gnrhr* in these organs will stimulate investigation of the underlying molecular mechanisms. Furthermore, expression in the salivary gland, branchia, and radula retractor muscle [47] suggest nonreproductive roles of oct-GnRHs.

The injection of ap-GnRH into sexually mature and immature sea hare (*A. californica*) for 10–14 days resulted in stimulation of the parapodial opening, inhibition of feeding, and promotion of substrate attachment but no reproduction-related functions. Specifically, there was no alteration in ovotestis mass, reproductive tract mass, egg-laying, penile eversion, oocyte growth, or egg-laying hormone accumulation and secretion [51]. These biological effects are at least partially in accordance with the expression of ap-GnRHR in the abdominal, cerebral, and buccal ganglia of the central nervous system, chemosensory organ, small hermaphroditic duct, and ovotestis [48,52]. Consequently, these findings suggest that ap-GnRH is not involved in reproductive functions in the sea hare, although the possibility cannot be excluded that the ap-GnRH injection experiment failed to reproduce in vivo functions in the genital organs.

In another gastropod, the abalone *Haliotis asinina*, treatment with 250 and 500-ng/g body weight (BW) synthetic Has-GnRH and buserelin (a synthetic mammalian GnRH analog) in vitro and in vivo enhanced oogonia and oocyte proliferation in one-year-old female *H. asinina* [40]. In addition, the injection of 1000-ng/g BW Has-GnRH and buserelin was less effective in stimulating oocyte cell proliferation [40], suggesting its involvement in feedback-like systems in abalone. Quite recently, Sharker et al. [53] verified the in vivo effects of Hdh-GnRH1 and -2 in the abalone *Haliotis discus hannai.* The injection of 250 and 500 ng/g BW of Hdh-GnRH1, -2, and buserelin resulted in the enhancement of spermatogonial cell and oogonial cell proliferation. Moreover, the expression of GnRH, GnRHR, and the serotonin receptor in the cognate male and female gonads were shown to be upregulated by the treatment with 250-ng/g BW Hdh-GnRH and buserelin [53]. These biological effects are compatible with the expression of the putative cognate receptor Hdh-GnRHR [46]. Similar to Has-GnRH in *Haliotis asinine* [40], the injection of 1000-ng/g BW Hdh-GnRH and buserelin was less effective than the aforementioned gonadal cell proliferation and gene expression effects in *Haliotis discus hannai* [53].

In the bivalve *Mizuhopecten* (*Patinopecten*) *yessoensis*, py-GnRH was found to enhance spermatogonial cell division in cultured testis [54]. Furthermore, a six-week administration of py-GnRH to yesso scallops using an original slow-release delivery system resulted in the acceleration of spermatogenesis and suppression of oocyte growth [55]. These biological functions are compatible with the high-level expression of the cognate receptor my-invGnRHR in the gonads during maturation [56]. Intriguingly, py-GnRH is highly likely to regulate masculinization. The administration of py-GnRH for six weeks to the scallop gonads altered the sex ratio (male:female = 1:1) to 100% male scallops by stimulating the generation of unnatural hermaphrodites (8.3% in week two and 33.3% in week four), whereas the sex ratio of the untreated group was unaffected [55]. These findings strongly suggest that py-GnRH is responsible for sexual differentiation or male gonad maturation. In particular, an enhancement of spermatogonial cells has been observed in mollusks but not in vertebrates, suggesting that this biological role is specific to mollusks or possibly other phyla of invertebrates. In combination, molluscan GnRHs play various unique roles in both reproductive and nonreproductive systems, with abalone and scallop GnRHs likely playing pivotal common roles in the proliferation and development of male germ cells. Exploration of the biological roles of GnRHs in various species will aid in verifying the functional evolutionary origin of ancestral GnRHs.

## 6. Conclusions and Perspectives

As stated above, a number of GnRH-GnRHR-signaling cascades remain to be investigated in invertebrates, despite a growing body of reports on the molecular cloning of invertebrate GnRHs and GnRHRs via omics analyses and in silico homology-searching. Nevertheless, the hitherto-elucidated GnRH-GnRHR signaling pathways in urochordates, echinoderms, and mollusks confirm that intracellular calcium mobilization is a major second messenger. In other words, the GnRHergic PLC-IP_3_ intracellular calcium ion pathway is conserved in invertebrates. This is in agreement with the view that GnRH-GnRHR signaling and CRZ-CRZ receptor signaling might have originated from common ancestors, given that, in addition to the moderate sequence similarity of CRZs and CRZ receptors to GnRHs and GnRHRs, respectively, intracellular calcium ion is also the second messenger in CRZ-CRZ receptor signaling pathways [7,9,10,11]. On the other hand, the functional characterization of GnRHRs has revealed that intracellular calcium mobilization is the major signaling pathway for GnRHRs in cephalochordates, echinoderms, and protostomes, whereas all *Ciona* GnRHR1 to -3 regulate the cAMP production pathways [9,12,13]. Notably, multiple GnRHs and GnRHRs have emerged in cephalochordates, urochordates, and vertebrates, whereas only a single GnRH and GnRHR have been identified in each echinoderm or protostome species (Table 1). Moreover, neither Gs nor Gi-coupled signaling has been detected in invertebrate GnRH-GnRHR signaling pathways, except for *Ciona* paralogs (see Section 1). These findings suggest that some invertebrate GnRHRs may also participate in the signaling of cAMP regulation. Elucidation of the functional interactions between more GnRHs and the cognate receptors in a wide range of invertebrate species will likely provide crucial clues to understanding molecular and functional divergence and conservation in the animal kingdom. Combined with the tissue distribution of mRNAs, the identification of ligand–receptor pairs is a crucial step in elucidating the endogenous roles of GnRHs. Of particular interest are the biological roles of GnRHs in protostomes, echinoderms, cephalochordates, and urochordates, all of which lack the HPG axis. As illustrated in Figure 3, GnRHs exhibit both common and species-specific reproductive and nonreproductive functions. Invertebrate GnRHs are highly likely to act directly on the genital organs in a neuropeptidergic manner, as supported by the expression of cognate GnRHRs in the ovary, follicles, and testis. In particular, GnRHs were shown to induce the proliferation of spermatogonial cells in several mollusks (Figure 3), suggesting a central role of GnRHs in the growth and maturation of the male gonads.

In addition to the elucidation of ligand-receptor pairs, two research strategies will prompt advances in verifying GnRH signaling and its relevant biological functions. First is the development of artificial agonists and antagonists specific to GnRHRs in each species. In general, the exploration of a specific agonist and antagonist for a receptor is costly, time-consuming, and serendipitous. However, recently, Shiraishi et al. efficiently and systematically identified multiple species-specific novel neuropeptide–GPCR pairs of *C. intestinalis* Type A by a combination of machine learning-based prediction and cell-based functional characterization. This strategy can also be applied to the prediction of artificial peptidic agonist and antagonist candidates in various organisms [57], likely supporting the systematic and efficient development of a species-specific agonist and antagonist for invertebrate GnRHRs. Second is the development of genome editing for invertebrates. Invertebrate genome editing experimental techniques, to date, have been developed only in insects, nematodes, and *Ciona* [58]. Thus, the development of genome editing in other invertebrates—in particular, starfish and mollusks—is highly likely to aid in the elucidation of the biological roles of GnRHs in invertebrates, given that the cognate GnRH-GnRHR pairs have been elucidated (Table 1) and several in vivo biological functions have been observed in these species (Figure 3). The identification of authentic GnRH–GnRHR pairs, combined with the development of species-specific GnRH agonists and antagonists and gene editing techniques for each invertebrate species, will pave the way for exploring the biological significance of invertebrate GnRHs, the underlying signaling mechanisms, and the evolutionary processes of biological functions that cannot be verified by a simple comparison of the sequence alignments and analyses of molecular phylogenetic trees and gene syntenies of GnRHs or GnRHRs.

## Figures and Tables

**Figure 1 ijms-21-08544-f001:**
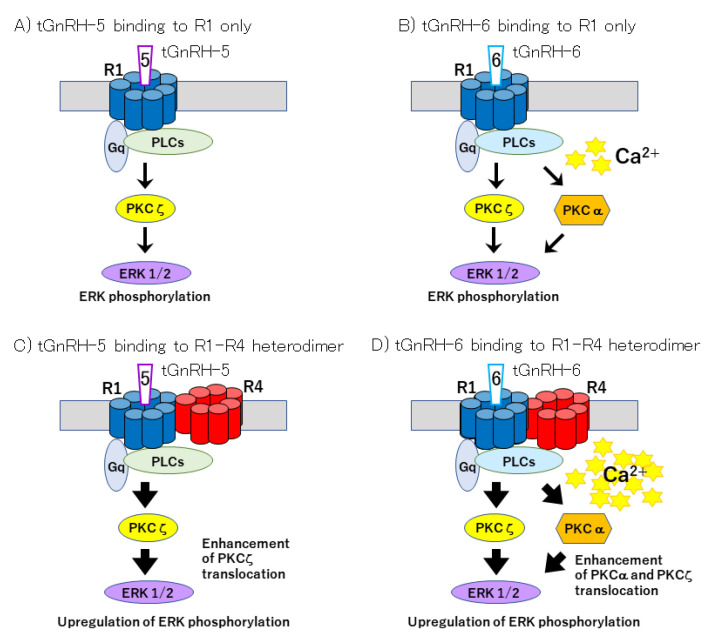
Signaling cascades triggered by the interaction of *Ciona* gonadotropin-releasing hormones (GnRHs) with the cognate receptor. (**A**) Interaction of tGnRH-5 with Ci-GnRHR-1 (R1) activates a calcium-independent protein kinase C (PKC), PKCζ, followed by the upregulation of extracellular signal-regulated protein kinase (ERK) phosphorylation. (**B**) Interaction of tGnRH-6 with R1 activates PKCζ and a calcium-dependent PKC, PKC α, leading to the upregulation of ERK phosphorylation. Both signaling pathways are potentiated via the heterodimerization of Ci-GnRHR-1 with a species-specific orphan GPCR paralog, Ci-GnRHR-4 (**C**,**D**).

**Figure 2 ijms-21-08544-f002:**
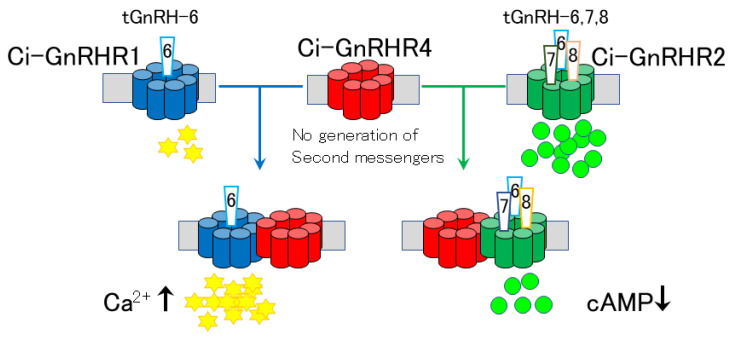
Differential regulation of GnRH signaling pathways via GPCR heterodimerization. Heterodimerization between Ci-GnRHR-1 and -4 results in a 10-fold more potent intracellular calcium ion mobilization response to tGnRH-6 compared with the Ci-GnRHR-1 monomer/homodimer. Heterodimerization between Ci-GnRHR-2 and -4 decreases cAMP production by 50% in response to tGnRH-6, 7, and 8 compared with the Ci-GnRHR-1 monomer/homodimer.

**Figure 3 ijms-21-08544-f003:**
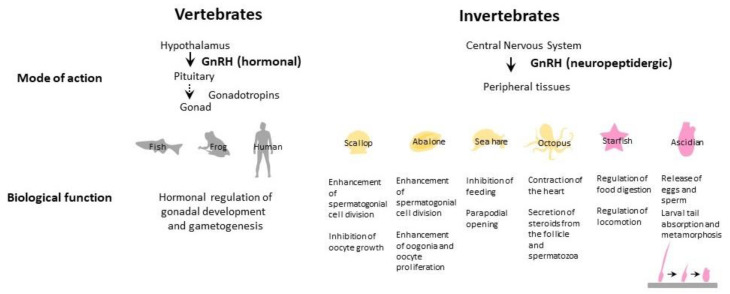
Major biological functions of GnRHs.

**Table 1 ijms-21-08544-t001:** Gonadotropin-releasing hormone (GnRH) signaling systems in the animal kingdom.

GnRH				
**Vertebrata**				
Species	Scientific Name	Peptide Name	Amino Acid Sequence	Receptors and Signaling Pathway
Human	*Homo sapiens*	GnRH1	pQ--HWSYGLRPGa	GnRHR1 (Ca^2+^)
		GnRH2	pQ--HWSHGWYPGa	Non-functional?
Guinea pig	*Cavia porcellus*	GnRH1	pQ--YWSYGVRPGa	GnRHR1 (Ca^2+^)
Sea bass	*Dicentrarchus labrax*	GnRH3	pQ--HWSYGWLPGa	dlGnRHR-II-1b (Ca^2+^, cAMP)
Lamprey	*Petromyzon marinus*	l-GnRH-I	pQ--HYSLEWKPGa	lGnRH-R-1 (Ca^2+^, cAMP), -3(Ca^2+^)
		l-GnRH-II	pQ--HWSHGWFPGa	lGnRH-R-1 (Ca^2+^, cAMP)
		l-GnRH-III	pQ--HWSHDWKPGa	lGnRH-R-1 (Ca^2+^, cAMP), -2(Ca^2+^)
**Urochordata**				
Tunicate	*Chelyosoma productum*	tGnRH-1	pQ--HWSDYFKPGa	N.D.
		tGnRH-2	pQ--HWSLCHAPGa	N.D.
		tGnRH-3	pQ--HWSYEFMPGa	Ci-GnRHR-3 (cAMP)
	*Ciona intestinalis*	tGnRH-4	pQ--HWSNQLTPGa	Ci-GnRHR-2 (cAMP)
		tGnRH-5	pQ--HWSYEYMPGa	Ci-GnRHR-3 (cAMP)
		tGnRH-6	pQ--HWSKGYSPGa	Ci-GnRHR-1 (Ca^2+^, cAMP), -2 (cAMP)
		tGnRH-7	pQ--HWSYALSPGa	Ci-GnRHR-2 (cAMP)
		tGnRH-8	pQ--HWSLALSPGa	Ci-GnRHR-2 (cAMP)
	*Ciona savignyi*	tGnRH-9	pQ--HWSNKLAPGa	N.D.
	*Ciona intestinalis*	Ci-GnRH-X	pQ—HWSNWWIPGAP	Partial antagonist for Ci-GnRHR-1, 3
			GYNGa	
	*Halocynthia roretzi*	tGnRH-10	pQ--HWSYGFSPGa	N.D.
		tGnRH-11	pQ--HWSYGFLPGa	N.D.
**Cephalochordata**				
Amphioxus	*Branchiostoma floridae*	Amph.GnRHv	pQE-HWQYGHWYa	unidentified
		Amph.GnRH-like	pQILCARAFTYTHTWa	Bf-GnRHR-3 (Ca^2+^)
**Echinodermata**				
Sea urchin	*Strongylocentrotus purpuratus*	Sp-GnRHP	pQVHHRFSGWRPGa	N.D.
Starfish	*Asterias rubens*	Ar-GnRH	pQIHYKNPGWGPGa	ArGnRHR (Ca^2+^)
**Protostomia**				
**Mollusca**				
Octopus	*Octopus vulgaris*	oct-GnRH	pQNYHFSNGWHPGa	Oct-GnRHR (Ca^2+^)
Cuttlefish	*Sepia officinalis*	oct-GnRH	pQNYHFSNGWHPGa	N.D.
Swordtip squid	*Loligo edulis*	oct-GnRH	pQNYHFSNGWHPGa	N.D.
Oyster	*Crassostrea gigas*	Cg-GnRH	pQNYHFSNGWQPa	N.D.
Yesso scallop	*Mizuhopecten (Patinopecten) yessoensis*	my-invGnRH (py-GnRH)	pQNFHYSNGWQPa	my-invGnRHR (Ca^2+^)
Sea hare	*Aplysia californica*	ap-GnRH	pQNYHFSNGWYAa	ap-GnRHR (Ca^2+^)
Owl limpet	*Lottia gigantean*	Lg-GnRH	pQHYHFSNGWKSa	N.D.
Giant triton snail	*Charonia tritonis*	Ctr-GnRH	pQNYHYSNGWHPa	N.D.
Abalone	*Haliotis asinina*	Has-GnRH	pQNYHFSNGWYPGa	N.D.
	*Haliotis laevigata*	Hlae-GnRH	pQNYHFSNGWHAa	N.D.
Pacific abalone	*Haliotis discus hannai*	Hdh-GnRH	pQNYHFSNGWYAa	Hdh-GnRHR (No functional assay)
**Annelida**				
Marine worm	*Capitella teleta*	Ca-GnRH	pQAYHFSHGWFPa	N.D.
Leech	*Helobdella robusta*	Hr-GnRH	pQSIHFSRSWQPa	N.D.
**Crustacea**				
Mitten crab	*Eriocheir Sinensis*	N.D.	N.D.	Es-GnRHR (transcriptome)
Black tiger shrimp	*Penaeus monodon*	Pm-GnRH	(transcriptome)	Pm-GnRHR (transcriptome)

Notes: The *N*-terminal pyroglutamic acid and *C*-terminal amide are shown by “pQ” and “a”, respectively. N.D. indicates no data.

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
