# Peer review of "Invertebrate Gonadotropin-Releasing Hormone Receptor Signaling and Its Relevant Biological Actions"

_ijms, 2020, doi:10.3390/ijms21228544_

Round 1

Reviewer 1 Report

The manuscript: Invertebrate Gonadotropin-Releasing Hormone Receptor Signaling and its Relevant Biological Actions by Sakai et al, gives a concise, informative and systematic overview of our knowledge on invertebrate GnRH-GnRHR signaling. The manuscript is very well written and interesting to read and thus, I have only minor corrections/suggestions for the authors.

The authors have contributed greatly to the knowledge on GnRH-GnRHR signaling in invertebrate species and to the research reviewed here. Therefore, they are entitled to be less modest about it and replace Sakai et al. (or similar references) with: our group showed, we discovered, etc.

Line 42- “AG protein”-I believe G is sufficient

Line 100- Given that sequencing of Ciona genome is almost complete are the genes coding for GnRHRs identified?

Line 112 - I would replace “the” with “these” (to implicate the cell lines mentioned in the previous sentence)

Line 264-265- This sentence does not feel complete.

Line 331- Remove underline

Line 358- I believe that the reference cited here should be: Jung LH, Kavanaugh SI, Sun B, Tsai PS. Localization of a molluscan gonadotropin-releasing hormone in Aplysia californica by in situ hybridization and immunocytochemistry. Gen Comp Endocrinol. 2014 Jan 1;195:132-7. doi: 10.1016/j.ygcen.2013.11.007. Epub 2013 Nov 15. PMID: 24246309. Please re-check all references.

Line 408- I see the dilemma. My suggestion (not necessarily the best solution): Notably, multiple GnRHs and GnRHRs have emerged in cephalochordates, urochordates, and vertebrates, whereas only a single GnRH/GnRHR has been identified in each up to now studied echinoderm or protostome species. 

Line 429- In different invertebrate species?

Fig. 3. Although the pituitary and gonadotropins are presented in the figure, less informed reader may still be misled to think that gametogenesis and gonadal development are directly regulated by GnRH. “Regulation (or control) of reproductive functions” may work better for this figure.

Author Response

Point-by-point responses to comments.

Comments of reviewers are displayed in italics. Our responses follow each comment.

<Response to editors>

We have included funding information in the Acknowledgements section.

Responses to reviewers

[To Reviewer #1]

The manuscript: Invertebrate Gonadotropin-Releasing Hormone Receptor Signaling and its Relevant Biological Actions by Sakai et al, gives a concise, informative and systematic overview of our knowledge on invertebrate GnRH-GnRHR signaling. The manuscript is very well written and interesting to read and thus, I have only minor corrections/suggestions for the authors.

The authors have contributed greatly to the knowledge on GnRH-GnRHR signaling in invertebrate species and to the research reviewed here. Therefore, they are entitled to be less modest about it and replace Sakai et al. (or similar references) with: our group showed, we discovered, etc.

<Response>

Thank you very much for your preferable and constructive suggestions. According to the suggestion, we have replaced “Sakai et al.” with “we showed (Line 123)”, “our group demonstrated (Line 204)”.

Line 42- “AG protein”-I believe G is sufficient

<Response>

We have replaced “G protein” with “G”.

Line 100- Given that sequencing of Ciona genome is almost complete are the genes coding for GnRHRs identified?

<Response>

In a previous study (Tello et al. Endocrinology, 2005), the authors stated that all putative Ciona GnRHR genes were detected and cloned by them.

Line 112 - I would replace “the” with “these” (to implicate the cell lines mentioned in the previous sentence)

<Response>

According to the suggestion, we have replaced “the” with “these”.

Line 264-265- This sentence does not feel complete.

<Response>

We have rephrased the sentence as follows:

“…it has been presumed that echinoderm GnRHs are included in the GnRH/CRZ family.”

Line 331- Remove underline

<Response>

We have removed it.

Line 358- I believe that the reference cited here should be: Jung LH, Kavanaugh SI, Sun B, Tsai PS. Localization of a molluscan gonadotropin-releasing hormone in Aplysia californica by in situ hybridization and immunocytochemistry. Gen Comp Endocrinol. 2014 Jan 1;195:132-7. doi: 10.1016/j.ygcen.2013.11.007. Epub 2013 Nov 15. PMID: 24246309. Please re-check all references.

<Response>

Thank you very much for the suggestion. We have added the above paper as reference 52. Moreover, we have reconfirmed all references. The following reference numbers have been changed according to the addition of the new reference.

Line 408- I see the dilemma. My suggestion (not necessarily the best solution): Notably, multiple GnRHs and GnRHRs have emerged in cephalochordates, urochordates, and vertebrates, whereas only a single GnRH/GnRHR has been identified in each up to now studied echinoderm or protostome species. 

<Response>

We agree with your suggestion, and have stated as such. We apologize for confusing you due to the remaining comments by our English-editing service.

Line 429- In different invertebrate species?

<Response>

We have replaced it with “GnRHRs in each species.”

Fig. 3. Although the pituitary and gonadotropins are presented in the figure, less informed reader may still be misled to think that gametogenesis and gonadal development are directly regulated by GnRH. “Regulation (or control) of reproductive functions” may work better for this figure.

<Response>

Thank you very much for the suggestion. According to your suggestion, we have rephrased the sentence as below:

“Hormonal regulation of gonadal development and gametogenesis”

Reviewer 2 Report

The manuscript is interesting and clearly written. It should be noted that part of the authors' research concerns precisely the subject of the manuscript.
The authors should justify why the Phyla Annelida and Crustacean are excluded from the review, of which data are known in the literature.
It is also required to adapt English to the UK or US form.

Author Response

Point-by-point responses to comments.

Comments of reviewers are displayed in italics. Our responses follow each comment.

[To Reviewer #2]

The manuscript is interesting and clearly written. It should be noted that part of the authors' research concerns precisely the subject of the manuscript.
<Response>

Thank you very much for your preferable and constructive suggestions.

The authors should justify why the Phyla Annelida and Crustacean are excluded from the review, of which data are known in the literature.

<Response>

By our literature-searching (PubMed, Google scholar, …), we cannot find papers showing elucidation of annelid or crustacean GnRH-GnRHR signaling or biological (endogenous) functions of GnRH of these species. I wonder if we missed the relevant papers, and would be grateful if you could kindly give us information about papers describing annelid or crustacean GnRH-GnRHR signaling or biological (endogenous) functions.

It is also required to adapt English to the UK or US form.

<Response>

Thank you for your suggestion. Our manuscript has been already edited and corrected by an English-editing service (Forte co. Ltd.: https://www.fortescience.com/). We would be grateful if you could kindly indicate incorrect (non-US or UK) forms in the manuscript. We will forward your opinions to the English-editing service.

Round 2

Reviewer 2 Report

The authors have provided adequate responses to my recommendations, therefore, the manuscript can be considered for publication on IJMS